# MSynFD: Multi-hop Syntax aware Fake News Detection

## ABSTRACT

The proliferation of social media platforms has fueled the rapid dissemination of fake news, posing threats to our real-life society. Existing methods use multimodal data or contextual information to enhance the detection of fake news by analyzing news content and/or its social context. However, these methods often overlook essential textual news content (articles) and heavily rely on sequential modeling and global attention to extract semantic information. These existing methods fail to handle the complex, subtle twists[1] in news articles, such as syntax-semantics mismatches and prior biases, leading to lower performance and potential failure when modalities or social context are missing. To bridge these significant gaps, we propose a novel **m**ulti-hop **s**yntax aware **f**ake **n**ews **d**etection (MSynFD) method, which incorporates complementary syntax information to deal with subtle twists in fake news. Specifically, we introduce a syntactical dependency graph and design a multi-hop subgraph aggregation mechanism to capture multi-hop syntax. It extends the effect of word perception, leading to effective noise filtering and adjacent relation enhancement. Subsequently, a sequential relative position-aware Transformer is designed to capture the sequential information, together with an elaborate keyword debiasing module to mitigate the prior bias. Extensive experimental results on two public benchmark datasets verify the effectiveness and superior performance of our proposed MSynFD over state-of-the-art detection models.

## CCS CONCEPTS

• **Computing methodologies** → **Artificial intelligence; Natural language processing**.

## KEYWORDS

Fake News Detection, Graph Neural Network, Debiasing

**ACM Reference Format:**
. 2018. MSynFD: Multi-hop Syntax aware Fake News Detection. In *Proceedings of Make sure to enter the correct conference title from your rights confirmation emai (Conference acronym 'XX)*. ACM, New York, NY, USA, 10 pages. https://doi.org/XXXXXXX.XXXXXXX

## 1 INTRODUCTION

The explosion of news consumption and sharing on social media platforms has created an unprecedented environment for the rapid dissemination of fake news. With the ease and speed at which

---

[1]A "subtle twist" refers to a slight, inconspicuous, or nuanced change or alteration that is unexpected and not immediately apparent.

information can be shared online, false narratives and misleading content can quickly gain attraction and reach a wide range of audiences. This proliferation of fake news poses a significant risk to society as it has the potential to manipulate public opinions, distort facts, and undermine trust in credible sources of information [12]. Recognizing this issue, there is a growing recognition of the urgent need to address the challenge of detecting fake news [52].

With the impressive advancements in deep learning, deep neural networks have gained widespread adoption in fake news detection in recent years. Various advanced neural models have been explored for fake news detection, including Recurrent Neural Networks (RNN) [17], Convolutional Neural Networks (CNN) [41, 49], attention networks [24, 48], and Graph Neural Networks (GNN) [35, 52]. These models leverage news texts or visual content and contextual information to identify the distinguishing features of fake news, yielding impressive detection performance. While the integration of multimodal information and social context has proven beneficial for detecting fake news, approaches relying heavily on visual and contextual cues suffer from the absence of such modalities or context, thus limiting their practicality in real-life scenarios. Consequently, text-based approaches have attracted significant attention as they primarily rely on news text, serving as the most crucial source of information in various fake news detection models.

Prevalent text-based detection approaches primarily revolve around RNN-based [8, 34], CNN-based [20, 26], and attention-based methods [9, 34, 48], which are inclined to capture comprehensive semantic correlations. However, these existing methods often lead to the acquisition of irrelevant information or word associations, presenting limitations when detecting fake news with subtle twists. Such kind of fake news articles often contain mostly true information but introduce false details through slight reversals or comparisons. As illustrated in Figure 1 (a), since most of the news content is about India, it may be misleading that 'our' refers to 'India', which causes the misunderstanding of the entire news segment. Such syntax-semantics mismatch, e.g., referential transfer, easily deceives the aforementioned semantic-targeted models, leading to inferior detection performance.

Additionally, it is crucial to address the presence of prior biases towards specific words, which has often been overlooked in previous methods. These biases arise from the statistical tendencies of neural models towards historical data and can result in an unfair viewpoint [43, 54], leading to misclassification of news articles, particularly those containing fake news [10]. Figure 1 (b) illustrates this issue, where preconceived notions about the emotional word "shock" and the entity word "India" can easily influence interpretation and judgment, potentially leading to the misidentification of genuine news as fake news. Zhu et al. [54] first introduced causal learning to mitigate entity bias in fake news detection, explicitly improving the generalization ability of detectors to future news data. However, we recognize that these prior biases primarily originate not only from key entities in news articles but also from significant contextual indicators such as emotional words like "shocks" in Figure 1(b). Since fake news often exhibits distinctive writing

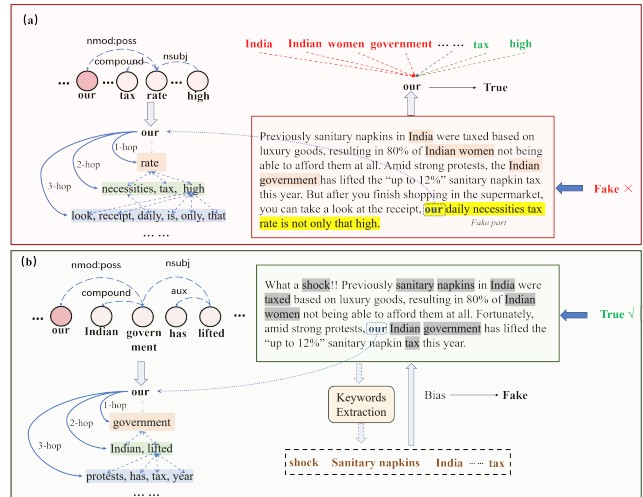

**Figure 1: (a) A fake news example with misleading information is highlighted in yellow. The word correlations above show how irrelevant words affect the understanding of the center word 'our', then mislead the detection result; (b) A true news example including keywords marked in grey and words leading to potential prior bias list below. The left region of both (a) and (b) shows syntax-associated words towards the center word 'our' at the 3-hops case and the local structure of the syntactic dependency tree.**

styles [55], characterized by exaggeration or extreme stances, it becomes imperative to adaptively learn and mitigate biases towards specific words rather than focusing on entity words.

To tackle the aforementioned challenges, a practical solution is to incorporate a syntactical dependency graph as supplementary information to enhance semantic learning and facilitate debiasing. However, modeling such syntactical dependency graphs presents three critical issues that need to be tackled: 1) Insufficient information from adjacent perception: The structure of adjacent perception may not provide enough contextual information. 2) Noisy information from imperfect parsing performance: Imperfect parsing can introduce noisy information into the syntactical dependency graph. 3) Lack of sequential information in syntactical dependency graphs [33]: Syntactical dependency graphs inherently lack sequential information. These issues pose significant challenges when it comes to effectively incorporating syntax analysis to address syntax-semantics mismatch and mitigate prior biases.

In light of the above discussion, we present a novel approach called Multi-hop Syntax aware Fake News Detection (MSynFD) that leverages the information provided by a syntactical dependency graph among news pieces. To address the limited perception range, we introduce the Subgraph Aggregation Attention (SAA) module. The module employs a syntactical multi-hop subgraph aggregation mechanism to extend the perception range of words, enabling capturing more comprehensive information about hierarchical syntactic structures. To tackle noisy information, we incorporate an adaptive gating mechanism into the SAA module to filter out noisy structural information, maintaining more relevant and reliable information. Recognizing the reliability of direct relations,

we further introduce a graph relative position bias mechanism that emphasizes the significance of low-hop relations. Furthermore, to tackle the lack of sequential information, we devise a sequential relative position-aware Transformer to capture sequential information for complementing the syntactical dependency graph. Our proposed Transformer seamlessly integrates with the SAA module, improving the interpretation and detection of fake news. Extensive experiments on public datasets verify the effectiveness and state-of-the-art performance of our detection method.

The main contributions of this paper are as follows:

- We propose a novel multi-hop syntax-aware fake news detection model, named MSynFD, to deal with fake news with subtle twists, effectively tackling syntax-semantics mismatch and mitigating prior biases in news articles.
- We design a multi-hop subgraph aggregation mechanism to capture comprehensive syntactic information, seamlessly integrating with a relative position-aware Transformer.
- We design a keywords-based debiasing to mitigate the pre-conceived notion within the news piece.

## 2 RELATED WORK

### 2.1 Fake News Detection

Fake news detection is conventionally framed as a binary classification task. This task can be broadly categorized into two main approaches: social-context-based and content-based [30].

**Social-Context-Based Detection:** Social-context-based methods revolve around the dynamics of news dissemination. Representative methods include 1) News dissemination-based approaches, which use GNN-based methods to model social interactions between users, news, and media sources [21, 31, 42, 52]; 2) User credibility-based approaches, which prioritize assessing the credibility of users and news sources in the context of fake news dissemination [1, 14]; 3) Feedback-based approaches, which rely on the user actions, e.g., comment [28, 53] and preference [5, 40].

**Content-Based Detection:** Content-based methods are grounded in the analysis of news content, incorporating text, visuals, and additional information to detect fake news. In the early stages, this analysis primarily relied on manual extraction of content, thematic elements, and user-related information, Detection techniques included machine learning models, including Decision Tree [2]and SVM [46]. More recently, deep learning models have achieved exceptional performance in the detection of fake news across various forms, including both unimodal text and textual-graphical multimodal data. For instance, RNN-based [8, 17, 18] methods leverage the sequential nature of textual data, while CNN-based [20, 41, 49] methods borrow from convolution concepts in computer vision to extract textual features. Attention-based [18, 24, 34, 39, 48] methods, which are particularly popular, utilizing attention mechanism [37] to capture relations within or between text from a global perspective. GNN-based methods focus on textual graph construction within documents[36] or the syntactical dependency relation between words [16, 32] Additionally, methods using external factual verification [13, 50] have contributed to enhanced detection performance.

Both content-based and social-context-based approaches necessitate effective text content modeling for node encoding. Moreover, since irrelevant connections caused by RNN-based, CNN-based,

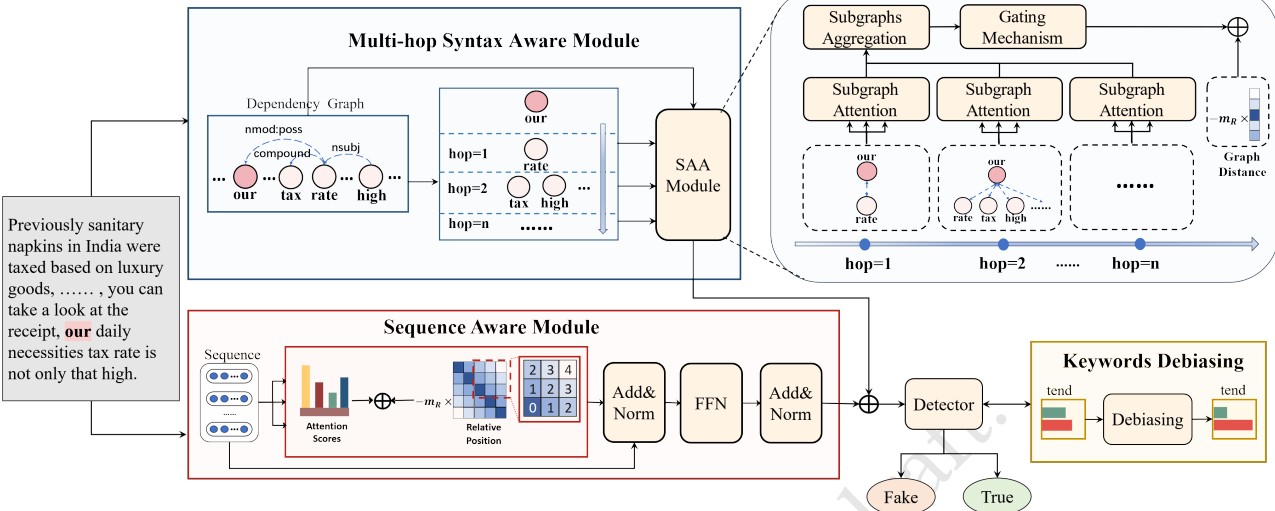

**Figure 2: Overview of our MSynFD fake news detection method.**

and Attention-based methods could bring noisy information, syntactical dependency information should be considered introduced in text content modeling. While previous studies have leveraged syntactical dependency graphs, there remains a need for deeper exploration of these graphs to extract more syntactical relations and filter out noisy connections that may introduce irrelevant information. Besides, prior biases are another factor that needs to be considered, as they can impact the generalization capacity of fake news detection[54]. However, little research has been dedicated to understanding and mitigating such biases.

## 2.2 Graph Neural Networks

In the context of fake news detection, GNN-based methods are predominantly employed in social-context-based approaches for modeling news dissemination and interactions[21, 22, 42, 52]. Nevertheless, GNNs have also demonstrated success in modeling textual content based on syntactical dependency graphs. These approaches typically entail using GNN-based methods, like GCN [11, 33] and GAT [7, 38], to encode the syntax graph predicted by off-the-shelf dependency parsers, subsequently generating textual graph embeddings tailored to specific tasks, and more recent research focuses on synergizing semantic and syntactical components to complement semantic information[15, 16, 32, 44]. However, GNN-based approaches face limitations. Traditional GNNs struggle with information exchange between non-local neighborhoods when two-word nodes are not in proximity. This challenge arises because the number of layers constrains the traditional approach to message passing, and extending this to larger values leads to overfitting and the loss of critical information[45, 51]. Although strategies like expanding the syntactical dependency graph to a global relation graph [45] and employing the graph spatial encoding [47] have shown promise, they introduce new issues, including an influx of irrelevant information and a lack of perception regarding sub-connected statements. In response, we propose aggregating subgraphs from a global syntactical dependency graph, attempting

to enhance the scope of perceived word nodes while filtering out irrelevant information. To the best of our knowledge, this represents a novel contribution to fake news detection.

## 3 PROBLEM DEFINITION

With a news piece as input, our objective is to determine whether they are fake news based on its textual information. Specifically, each news piece $C = \{P, G, K, Y\}$ consists of the news text $P$ containing $n$ words $P = \{w_1, w_2, \cdots, w_n\}$. The syntactical dependency graph $G = (V, E)$ obtained by HanLP[2] and Stanford CoreNLP tools[3] for Chinese and English news respectively, where $V$ is the set of graph nodes corresponding to the words in $P$, and $E$ is the set of edges representing the syntactical dependency relations between words. The keywords $K$ are obtained by KeyBERT [6] containing $m$ words $K = \{k_1, k_2, \cdots, k_m\}$, and the ground-truth label $Y \in \{0, 1\}$, where 1 and 0 denote the news piece is fake or true. The purpose of the fake news detection is to predict whether the label $C$ is 1 or 0.

## 4 METHOD

In this section, we discuss each component of our proposed MSynFD method (as shown in Figure 2) in detail.

### 4.1 Input Encoding

For each news $P$ with $n$ words, i.e., $P = \{w_1, w_2, \cdots, w_n\}$, we feed it into BERT to obtain its representation $\widetilde{P} = \{\widetilde{w}_1, \widetilde{w}_2, \cdots, \widetilde{w}_n\}$. For each word $w_i$ with $m$ tokens $w_i = \{w_{sub1}, w_{sub2}, \cdots, w_{subm}\}$, we obtain its representation by summing the embeddings of its tokens.

### 4.2 Multi-hop Syntax Aware Module

We introduce the Subgraph Aggregation Attention (SAA) module. It consists of the syntactical multi-hop information aware mechanism and the adaptive gating mechanism and introduces the graph

---

[2]https://hanlp.hankcs.com/
[3]https://stanfordnlp.github.io/CoreNLP

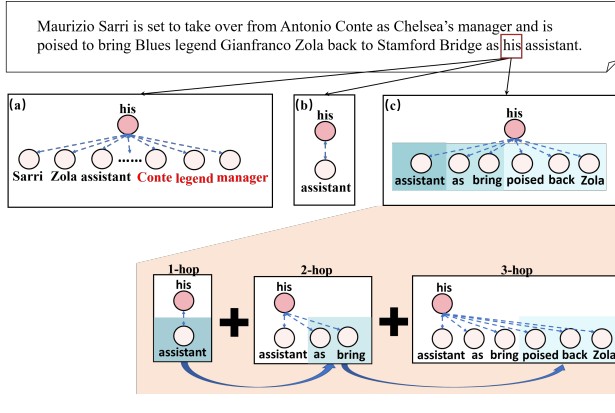

**Figure 3: Comparison illustration of information propagation among (a) attention-based methods, (b) traditional GNN-based methods, and (c) our Multi-hop Syntax aware module.**

relative position bias. These components collectively capture information between words from the syntactical perspective and, importantly, prevent the formation of irrelevant connections.

When considering a central word, such as "his", as illustrated in Figure 3, the global connection of the attention-based method makes a lot of irrelevant connections like "Conte" and "manager" and brings noisy information to "his". Meanwhile, the information from adjacent words of the traditional GNN-based method often provides insufficient information. For instance, we could know little about "assistant". To address these limitations, multi-hop information becomes crucial for a more accurate understanding. For example, we can ascertain that the "assistant" refers to "Zola" and that he has been "poised" within 3-hop syntactical dependency relations. Accordingly, we have introduced a syntactical multi-hop information-aware mechanism, allowing us to perceive interactions within a range of $m$-hop. Firstly, we obtain $m$ adjacent matrices, with $A^d \in R^{n \times n}$ representing the $d$-th hop subgraph from the syntactical dependency graph $G$. In these matrices, $A^d_{ij}$ is set to 1 if word $w_i$ can be reached from $w_j$ within $d$ words otherwise $A^d_{ij}$=0. And we set $A^d_{ii} = 1$ for the self-connection, then the adjacency matrix $\widetilde{A}^d$ can be updated to $\widetilde{A}^d = A^d + I$. Note that the adjacent matrix indicates whether two words have a relation instead of the strength of the relation with specific values. Considering the varying word relations derived from different interaction scenarios within specific hop subgraphs, we introduce a hop-specific subgraph attention mechanism to determine the hop-based relation value.

Initially, we transform the news representation $\widetilde{P}$ into word node features $H= \{h_1, h_2, \cdots, h_n\}$ by the linear transformation with trainable parameters $W_P$. To account for the dynamics of word relations under different connections, we employ the hop-specific trainable weight matrix $W^d_A$, which is used to parameterize every word node. This enables the calculation of an edge weight matrix $Z^d$ for the $d$-th hop subgraph, where the element $z^d_{ij}$ signifies the relation value between word node $i$ and word node $j$:

$$H = W_P \widetilde{P}$$
$$z^d_{ij} = LeakyReLU(W^d_A h_i, W^d_A h_j)\widetilde{A}^d_{ij} \quad (1)$$

As the perceived range expands, the potential for irrelevant and noisy information increases, diluting the special local information. To address this, we employ an adaptive adjustment mechanism to measure the importance of information from various subgraphs through a learnable parameter $W_Z$, allowing the model to balance the information from adjacent relation among subgraphs with varying hops. Denoting the set of multi-hop relation value $Z = [Z^1, Z^2, \cdots, Z^m]$, we have:

$$S = \sigma(W_Z)Z \quad (2)$$

where $\sigma$ is the sigmoid function. To capture and filter the noise, we introduce a gating mechanism using another learnable parameter $W_H$ to generate a gating matrix $M$. This is shared by the word nodes to discern and eliminate the noise, subsequently refreshing the value matrix as $S^{'} = MS$:

$$M = \begin{cases} 1 & if \quad v_{ij} > t_i \\ 0 & else \end{cases} \quad (3)$$
$$T = W_H H$$

where $T = [t_1, t_2, \cdots, t_n]$ is a set of adaptive thresholds to word nodes. Furthermore, shorter graph distances between two words indicate stronger relevance. Hence, we introduce a direct use of graph relative position from the global graph structure $G$, which is used as an attention bias added after the aggregation and filtering processes, enhancing adjacent attention between words within the syntactical structure during the softmax function-based attention calculation mechanism. The output graph representation $\widetilde{H} = \widetilde{S}H$, with $\widetilde{s}_{ij}$ in $\widetilde{S}$ can be defined as:

$$\widetilde{s}_{ij} = \frac{exp(s^{'}_{ij} - m_G|d_{ij}|)}{\sum_{k=1}^n exp(s^{'}_{ik} - m_G|d_{ik}|)} \quad (4)$$

where $d_{ij}$ represents the graph's relative distance between word nodes $i$ and $j$, $m_G$ stands for a head-specific fixed slope. With $h$ heads, the slopes are the geometric sequence: $\frac{1}{2^1}, \frac{1}{2^2}, \cdots, \frac{1}{2^h}$. We adjust the receptive field and filter the noise edges before using the graph relative position bias to ensure that only the relation between nodes' features is used to evaluate the reliability of information transmissions. To stabilize the learning process of the SAA module, the mechanism above is extended to the multi-head form with $h$ heads. After concatenating the outputs from each head, the ultimate graph representation can be obtained after a normalization layer:

$$\widetilde{H} = [\widetilde{H}^{(1)}, \widetilde{H}^{(2)}, \cdots, \widetilde{H}^{(h)}]$$
$$\widetilde{H} = Norm(\widetilde{H}) \quad (5)$$

## 4.3 Semantic Aware Module

Giving an input news representation $\widetilde{P}$, the information from syntactical structures may be limited, and potential syntactical errors might exist, so the transformer structure is employed to extract semantic information. The objective is to ensure that each word can obtain information from a global perspective while perceiving the sequential structure. Inspired by the textual positional embedding researches in recent years[23, 25], we introduce a sequential relative position bias, which can be added after query-key dot product to promote higher attention scores between adjacent words in a

sequence, leveraging the properties of softmax operator, to emphasize the stronger correlation among closer words. Specifically, for a transformer of multi-head design with $h$ heads, we obtain $Q^{(l)}, K^{(l)}, V^{(l)}$ on the $l$-th head as the query matrix, key matrix, and value matrix through three distinct linear transformations, and utilize $M_R$ as the sequential relative position matrix. As a result, the semantic representation on the $l$-th head $R^{(l)}$ can be defined:

$$Q = W_Q \widetilde{P}, \quad K = W_K \widetilde{P}$$
$$V = W_V \widetilde{P} + b_V$$
$$R^{(l)} = softmax(\frac{Q^{(l)}K^{(l)T}}{\sqrt{d}} - m_R^{(l)}M_R)V^{(l)} \tag{6}$$
$$r_{ij}^{(l)} = \frac{exp(q_i^{(l)}k_j^{(l)} - m_R^{(l)}|i-j|)}{\sum_{k=1}^{n} exp(q_i^{(l)}k_k^{(l)} - m_R^{(l)}|i-k|)}$$

where $W_Q$, $W_K$, $W_V$, $b_V$ are trainable parameters, $\sqrt{d}$ denotes the scaling factor. $m_R$ is another head-specific fixed slope, equal to $m_G$ in the experiments. We only introduce the trainable bias for $V^{(l)}$, which transforms the sequential relative position into a rigid bias, thereby encouraging the module to focus more on the sequential relation. After connecting the concatenated outputs from each head, a two-layer MLP is employed to extract higher-level semantic features, followed by two normalization layers and the residual structure. Thus, the final semantic representation is obtained:

$$R = [R^1, R^2, \cdots, R^h]$$
$$\widetilde{R} = Norm(R + \widetilde{P}) \tag{7}$$
$$\widetilde{R} = Norm(\widetilde{R} + FFN(\widetilde{R}))$$

### 4.4 Fake News Detector

For each news piece, we possess both the multi-hop graph representation $\widetilde{H}$ and the semantic representation $\widetilde{R}$. These two representations are then concatenated, yielding the fusion representation $\widetilde{F} = [\widetilde{R}, \widetilde{H}]$. Next, we use a sequence attention mechanism to gather information from each word:

$$F = \sum_{i=1}^{n} softmax(W_{Fi}\widetilde{f_i} + b_{Fi})\widetilde{f_i} \tag{8}$$

where $W_F$ and $b_F$ are trainable parameters. And in the end, we feed $F$ into a two-layer MLP to get the prediction $y'$:

$$y' = softmax(W_2(ReLU(W_1F + b_1)) + b_2) \tag{9}$$

where $W_1$, $W_2$, $b_1$, $b_2$ are trainable parameters.

### 4.5 Keywords Debiasing

We introduce a keywords debiasing module to mitigate prior bias from keywords. First, we train a simple keyword encoder with a pre-trained BERT to obtain prior keyword representation $K = \{k_1, k_2, \cdots, k_m\}$. Then, we use the maximum pooling to capture the most salient features of each keyword. Next, we train another classification layer to obtain the prediction from keywords $y_K$:

$$\widetilde{K} = BERT(K)$$
$$\widetilde{K}_{max} = Maxpool(\widetilde{K}) \tag{10}$$
$$y_K = softmax(W_4(ReLU(W_3\widetilde{K}_{max} + b_3)) + b_4)$$

**Table 1: Statistics of the datasets**

| Dataset | Weibo | | | GossipCop | | |
|---|---|---|---|---|---|---|
| | Train | Val | Test | Train | Val | Test |
| Fake | 2561 | 499 | 754 | 2024 | 604 | 601 |
| Real | 7660 | 1918 | 2957 | 5039 | 1774 | 1758 |
| Total | 10221 | 2417 | 3711 | 7063 | 2378 | 2359 |

where $W_3$, $W_3$, $b_3$, $b_4$ are trainable parameters.

For the training phase, the final prediction $\hat{y} = \alpha(y') + (1-\alpha)(y_K)$ fusion $y'$ and $y_K$ while $\alpha$ is a hyper-parameter to balance the two terms. We train the whole framework with the cross-entropy loss:

$$\mathcal{L}_O = \sum_{P,y\in\mathcal{D}} -ylog(\hat{y}) - (1-y)log(1-\hat{y})$$
$$\mathcal{L}_K = \sum_{P,y\in\mathcal{D}} -ylog(y_K) - (1-y)log(1-y_K) \tag{11}$$
$$\mathcal{L} = \mathcal{L}_O + \beta(\mathcal{L}_K)$$

where $\beta$ is another hyper-parameter to balance the two loss functions of fusion prediction and keywords-based prediction, and both $\alpha$ and $\beta$ are set as 0.1 in the experiments. This training procedure encourages the model to focus on and capture the prior keyword bias, allowing the fake news detector to learn less biased information. In the validation and test procedure, we only use $y'$ as the prediction of the model.

## 5 EXPERIMENTS

### 5.1 Datasets

We evaluate our MSynFD on two real-world datasets[4]. The Weibo dataset [27] ranging from 2010 to 2018[5] is used as the Chinese dataset, and the GossipCop data from FakeNewsNet [29][6] is used as the English dataset. Each news piece is labeled as fake or real in both datasets, and we only use the news content in the experiments. Besides, we keep the same dataset splitting as the organizers provide, where both datasets are segmented in chronological order to simulate real-world scenarios. Detailed statistics of both datasets used in our experiments are shown in Table 1.

### 5.2 Baselines

We choose nine content-based representative and/or state-of-the-art methods in fake news detection tasks for comparison, including RNN, CNN, GNN, attention, and debiasing models, and unimodal or multi-modal models. Since social-context-based methods focus on modeling information transmission and show high dependence on transmission structure, they are not included as baselines.

**Bi-GRU** [3] is an RNN-based model that uses a bidirectional GRU network to learn semantic associations within news.

**EANN** [41] is a multi-modal fake news detection model that uses TextCNN for text representation and use adversarial learning method to obtain the invariant features of news.

**BERT** [4] is a popular pre-training model used for fake news detection. We use the original BERT model for the GossipCop dataset and the Chinese version of BERT for the Weibo Dataset.

---
[4]https://github.com/ICTMCG/ENDEF-SIGIR2022
[5]https://github.com/ICTMCG/News-Environment-Perception/
[6]https://github.com/KaiDMML/FakeNewsNet

**Table 2: Fake news detection results on the Weibo dataset and the GossipCop dataset. The second best-performing methods are underlined, and ∗ indicates the statistically significant improvement (i.e., two-sided t-test with $p < 0.05$).**

| Method | Weibo | | | | | | GossipCop | | | | | |
|---|---|---|---|---|---|---|---|---|---|---|---|---|
| | Acc | macF1 | AUC | spAUC | $F1_{real}$ | $F1_{fake}$ | Acc | macF1 | AUC | spAUC | $F1_{real}$ | $F1_{fake}$ |
| BiGRU | 0.8214 | 0.7172 | 0.8354 | 0.6636 | 0.8887 | 0.5456 | 0.8379 | 0.7730 | 0.8634 | 0.7358 | 0.8943 | 0.6516 |
| EANN | 0.8197 | 0.7162 | 0.8276 | 0.6649 | 0.8875 | 0.5448 | 0.8517 | 0.7926 | 0.8765 | 0.7586 | 0.9033 | 0.6820 |
| BERT | 0.8474 | 0.7601 | 0.8754 | 0.7102 | 0.9048 | 0.6155 | 0.8439 | 0.7873 | 0.8781 | 0.7579 | 0.8968 | 0.6778 |
| MDFEND | 0.7786 | 0.7051 | 0.8301 | 0.6691 | 0.8519 | 0.5584 | 0.8518 | 0.7905 | 0.8712 | 0.7543 | 0.9037 | 0.6772 |
| HMCAN | 0.8289 | 0.7257 | 0.8300 | 0.6674 | 0.8939 | 0.5575 | 0.8490 | 0.7843 | 0.8479 | 0.7386 | 0.9025 | 0.6660 |
| BERT-Emo | 0.8438 | 0.7586 | 0.8743 | 0.7061 | 0.9019 | 0.6154 | 0.8455 | 0.7912 | 0.8800 | 0.7631 | 0.8974 | 0.6849 |
| BERT-Emo-ENDEF | 0.8584 | 0.7731 | 0.8838 | 0.7278 | 0.9121 | 0.6341 | 0.8520 | 0.8010 | 0.8855 | 0.7674 | 0.9020 | 0.6987 |
| CMMTN | 0.8706 | 0.7812 | 0.8723 | 0.7438 | 0.9211 | 0.6412 | 0.8593 | 0.8117 | 0.8889 | 0.7770 | 0.9064 | 0.7170 |
| MGIN-AG | 0.8666 | 0.7753 | **0.8959** | 0.7375 | 0.9185 | 0.6320 | 0.8593 | 0.8072 | 0.8916 | 0.7788 | 0.9074 | 0.7069 |
| **MSynFD** | **0.8787**∗ | **0.7889**∗ | 0.8903 | **0.7656**∗ | **0.9266**∗ | **0.6512**∗ | **0.8699**∗ | **0.8164**∗ | **0.8949**∗ | **0.7904**∗ | **0.9155**∗ | **0.7173** |

**MDFEND** [19] is a multi-domain-based fake news detection model integrating the Mixture of Experts(MOE) to capture the domain information of news.

**HMCAN** [24] is a multi-modal fake news detection model that designs a hierarchical encoding network to capture the rich hierarchical semantics text information of news.

**BERT-Emo** [53] is a fake news detection model that combines the emotional features of news content and social contexts.

**BERT-Emo-ENDEF** [54] is a fake news detection method that introduces an entity debiasing framework (ENDEF) in the BERT-Emo model to mitigate the bias within news pieces.

**CMMTN** [39] is a multi-modal fake news detection model that uses a masked Transformer to filter the noise or irrelevant context.

**MGIN-AG** [32] is a multi-modal rumor detection model that uses GCN to generate augmented features from claims, and attention mechanisms to extract the embedded text from images.

Since this work focuses on the textual content of news, all the multi-modal models are kept with their text-only version. For a fair comparison, the labels for the auxiliary event classification task of EANN and the domain labels of MDFEND are derived by clustering according to the publication year; BERT-Emo is a simplified version without the emotion in comments, and MGIN-AG does not use the embedded text in images but use the claim text itself as the replacement. While the results of Bi-GRU, EANN, BERT, MDFEND, BERT-Emo, and BERT-Emo-ENDEF would come from the [54], the remaining models will all use the same training parameters setting, and their classification results will be obtained by the same design of MLP classifier as our proposed MSynFD method, in which the activation function is ReLU and the dimension of hidden layer is set as 384. The heads of any multi-head structure are set to 12, and we report the average testing results over five runs.

## 5.3 Experimental Settings

Since the Weibo dataset and the GossipCop dataset have different average lengths, the maximum sequence lengths of the Weibo and GossipCop datasets are set to 150 and 350, respectively, and the batch size is 32. All models are implemented using PyTorch, and the Adam optimizer is used with a learning rate of 1e-5, and gradually decreases during training according to the decay rate of 1e-6. The hops of the syntactical dependency graph for the Weibo dataset and the GossipCop dataset are set as 4 and 3, respectively. We

use an early stop strategy for the label accuracy of the validation set, with a patience of 5 epochs. We adopt six metrics, including accuracy (Acc), macro F1 score (macF1), Area Under ROC (AUC), standardized partial AUC (spAUC), and the F1 scores of fake and real class ($F1_{fake}$ and $F1_{real}$) to evaluate detection performance. All the implementation codes will be publicly available for reproducibility once the review finished.

## 5.4 Performance Results

Table 2 shows the performance of all comparative methods on two public real-world datasets, where the best performance is marked in bold. Results show that our proposed MSynFD has achieved the best performance on five crucial metrics compared with the SOTA fake news detection models. On Weibo, MSynFD yields 0.81%, 0.77%, 2.18%, 0.55%, and 1.00% improvement, over Acc, macF1, spAUC, $F1_{fake}$ and $F1_{real}$, and over AUC is 0.56% lower than MGIN-AG model. Additionally, on GossipCop, MSynFD yields 1.06%, 0.47%, 0.33%, 1.16%, 0.81%, and 0.03% improvement, over Acc, macF1, AUC, spAUC, $F1_{fake}$ and $F1_{real}$. The results demonstrate that the proposed method can capture the local syntactical dependency structure information of news and mitigate the priori bias from keywords, which can help better understand and analyze the news piece.

The adversarial mechanism and the MOE may not be able to learn enough about fake news patterns in the short-text context, which causes EANN and MDFEND to perform well on the Gossip-Cop dataset but not on the Weibo dataset. Further, comparing the results between the HMCAN and CMMTN, the noisy irrelevant connections from the attention mechanism affect the model performance; with the help of the mask mechanism, CMMTN could perform better on both datasets.

Finally, the results of MGIN-AG show that the GNN model does play a role, making MGIN-AG perform better than BiGRU and HMCAN on both datasets. The results compared between BERT-Emo and BERT-Emo-ENDEF show that the debiasing framework does help improve model performance for fake news detection, providing a basis for rationalizing our design of MSynFD.

## 5.5 Ablation Study

To verify the effectiveness of the different modules of MSynFD, we compare them with the following variants:

**Table 3: Results of ablation study on both datasets**

| Method | Weibo | | GossipCop | |
|---|---|---|---|---|
| | Acc | macF1 | Acc | macF1 |
| MSynFD ¬ SR | 0.8739 | 0.7826 | 0.8618 | 0.8067 |
| MSynFD ¬ SAA | 0.8709 | 0.7792 | 0.8453 | 0.7656 |
| MSynFD ¬ KD | 0.8758 | **0.7938** | 0.8661 | 0.8121 |
| MSynFD-MH-GAT | 0.8717 | 0.7783 | 0.8512 | 0.8022 |
| **MSynFD** | **0.8787** | 0.7889 | **0.8699** | **0.8164** |

**MSynFD ¬ SR** removes the sequential representation module, which makes the model lose the ability to perceive the sequence position structure.

**MSynFD ¬ SAA** removes the subgraphs aggregation attention module, which makes the model lose the ability to perceive the local syntactical dependency structure.

**MSynFD ¬ KD** removes the keywords debiasing, which makes the model lose the ability to mitigate the priori bias from keywords within the news piece.

**MSynFD-MH-GAT** replaces the SAA module with GAT to validate its effectiveness in obtaining local syntactical dependency structure information. For a fair comparison, we adjust the traditional GAT to multi-hops(MH)-GAT, whose adjacency matrix is set to be the same hops adjacency case as the original model, to ensure both models capture structural information at the same depth.

Table 3 shows that when comparing MSynFD with MSynFD ¬ SR reduces the accuracy of the proposed model by 0.48% and 0.81%, and macro F1 score by 0.63% and 0.97% on the Weibo and the GossipCop datasets, respectively. This means that the sequential representation module helps complete the global sequential information and improves the performance of fake news detection.

Further, for MSynFD ¬ SAA reduces the accuracy of the proposed model by 0.78% and 2.46%, and macro F1 score by 0.97% and 5.08% on the Weibo and the GossipCop datasets, respectively. It means that the local syntactical dependency structure information focused by the SAA module can reduce the noisy information caused by irrelevant connections, which is reflected in the fact that model performance degrades much more in the long news dataset GossipCop than in the short text dataset Weibo.

For, MSynFD ¬ GAT reduces the accuracy by 0.70% and 1.87%, and macro F1 score by 1.06% and 1.42% on the Weibo and the GossipCop datasets, respectively. It means that though perceiving syntactical dependency structure at the same depth, the SAA module is more effective than GAT due to the subgraph weighted aggregation mechanism.

Finally, MSynFD ¬ KD reduces the accuracy of the proposed model by 0.29% and 0.38%, and macro F1 score by -0.49% and 0.43% on the Weibo and the GossipCop datasets, respectively. The results show that keyword bias can improve performance in some situations (see section 5.1 - qualitative analysis for details).

## 5.6 Qualitative Analysis

To explore how the size of the perceived range and keyword bias affect the performance of fake news detection. We designed a series of experiments about the number of syntactical dependency graph hops and the max length of a news piece. The results shown in Figure 4 (a) indicate that the performances of both the Weibo

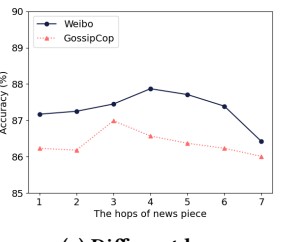
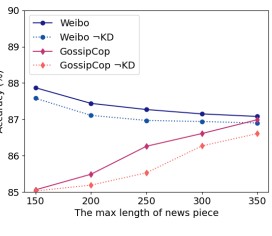

(a) Different hops                (b) Different max lengths

**Figure 4: (a) Performance of the MSynFD model under different values of the parameter hops; (b) Performance of the MSynFD model and MSynFD ¬ KD under different values of the parameter max length.**

dataset and the GossipCop dataset increase and then decrease as hops increase, which means that the perceived range in the local syntactical dependency graph has a certain threshold before reaching it, more effective information will be obtained, and after that, the irrelevant noise will be brought and reduces the performance. The best hops for these two datasets are 4 and 3, respectively.

As shown in Figure 4 (b), the effect of the keywords debiasing presents a different scenario. For the Weibo dataset, as the max length of the news piece increases, the performance improvement from the keywords debiasing becomes more insignificant. We think that this may be due to the average length of the Weibo dataset being 120, so limited information makes the bias within keywords as important information for detection, and with the max length increasing, the percentage of padding in the news piece increases and reduces information density, creates further reliance on bias information, and alleviates the effect of keywords debiasing; On the other hand, for the GossipCop dataset, the performance improvement from the keywords debiasing is increasing first from insignificant and decreasing a little. Since the average length of the GossipCop dataset is 606, we think at first the length of 150 lacks information, causing the bias within keywords, which is important for detection too. As the max length increases, the informative patterns grow, which alleviates the reliance on biased information, making the debiasing module more useful. With the max length increasing, more informative patterns are brought, and the effect of the keywords debiasing has been balanced.

## 5.7 Case Study

To provide an intuitive demonstration of the functions of each part, we use test set data from two datasets to analyze the intermediate process. We first test the performance of the Multi-hop Syntax Aware Module and Semantic Aware Module. As shown in Figure 5, due to the use of sequential relative position bias, the focuses of sequential neighbors are significantly enhanced in Chinese news, especially in Figure 5 (a), while it does not work well in English news. This may be from the grammatical differences between Chinese and English. And, the distant irrelevant connection, like 'Iceland' to 'China' in Figure 5 (b) and 'we' to 'fashion' in Figure 5 (c), would still be built. The SAA module does show the ability to avoid such irrelevant information while obtaining enough useful information. As shown in Figure 5 (c), the perception range is extended from the adjacent word "know" to the 3-hop adjacent word "Hadid". However, the hazard of information gaps still exists, as shown in Figure 5 (d);

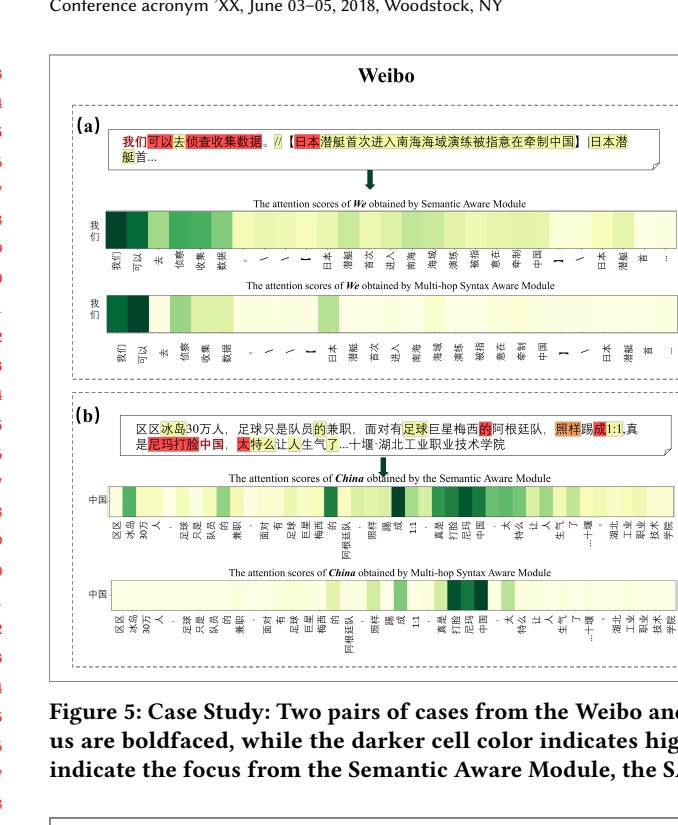

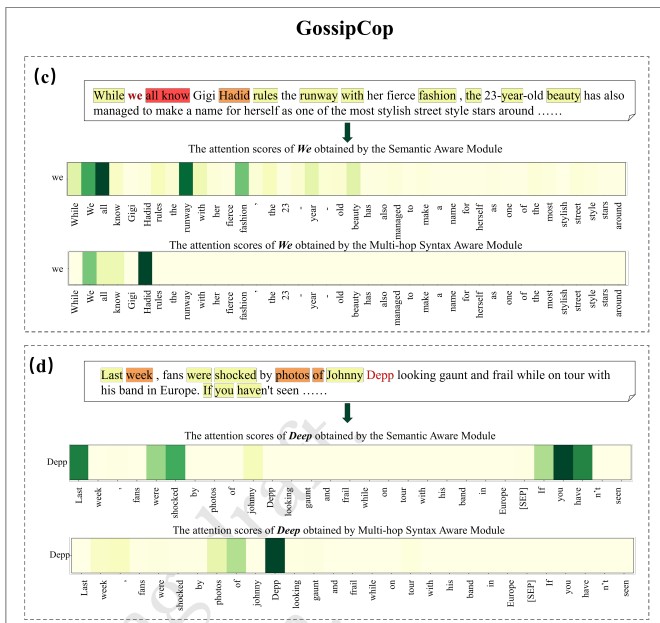

**Figure 5: Case Study: Two pairs of cases from the Weibo and the GossipCop datasets, respectively. The center words focused by us are boldfaced, while the darker cell color indicates higher attention value, the yellow areas, orange areas, and red areas indicate the focus from the Semantic Aware Module, the SAA Module, or both focus.**

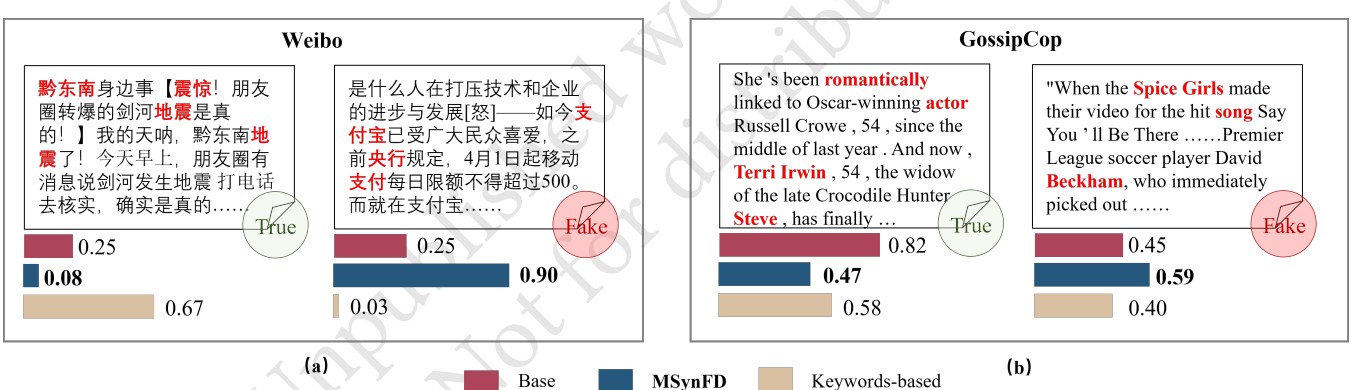

**Figure 6: Case Study 2: Two pairs of cases from the Weibo and the GossipCop datasets respectively. For each dataset, one case is true while the other is fake. The keywords are boldfaced, and the lengths of bars represent the probability predictions of fake news of our base model (keywords debiasing ablation model), our MSynFD method, and the keywords-based model.**

we cannot obtain how the photos are due to the limits of syntactical relations. So, the semantic complement is still necessary.

Then we analyze the distribution of prediction scores of our main model ablation keywords debiasing before and after, as Figure 6 shows, the keywords debiasing can mitigate the effect of words with prejudice (e,g, 'shock' in Figure 6 (a)) and words of authority (e.g. central bank in Figure 6 (a)). Although the keywords debiasing shows the ability to capture some non-entity keywords (e.g. 'shock', 'pay' in Figure 6 (a) and 'romantically' in Figure 6. (b)), it may ignore some important words that lead to misjudgment like 'Russell Crowe' due to the limits of Semantic-based keywords extraction method. Expanding the captured keywords is where our future research will focus on improvement.

## 6 CONCLUSION

In this paper, we propose a new fake news detection method, MSynFD, which uses a Multi-hop Syntax Aware Module to capture multi-hops syntactical dependency information within news pieces to extend the local syntax information of each word. Then, the Semantic Aware Module is used to obtain sequential aware semantic information. In the end, the Keywords Debiasing is mitigated into the model to mitigate prior bias from keywords. The experimental results have shown that among the state-of-the-art methods, our proposed MSynFD method achieves the SOTA performance. For future work, we plan to leverage multi-modal information to filter irrelevant information and enhance syntactical information.

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
