# OpenReview forum: "MSynFD: Multi-hop Syntax aware Fake News Detection"
_ACM.org/TheWebConf/2024/Conference — TheWebConf24_

### Official Review · Reviewer_c63p · 2023-11-23

**Novelty:** 6
**Technical Quality:** 5

**Review:**

This research utilizes a graph attention model and a dependency tree for text embedding. This embedding is then merged with another obtained from a transformer-based model, aiming to identify fake news. The effectiveness of this approach is demonstrated through its performance across six evaluation metrics and two datasets, highlighting the framework's superiority.

Key Strengths:
- Incorporating a dependency tree into the graph attention models enhances the focus on significant words and reduces noise, as compared to relying on the text's original sequential order. This approach is innovative.
- The paper's illustrations effectively convey the authors' concepts, aiding in the clarity and comprehension of their work.
- The writing is coherent and well-structured, particularly in the sections introducing concepts, which facilitates easy understanding of the paper.

**Questions:**

- A news may contain multiple sentences, how could your framework use the dependency tree to represent this news?
- What criteria are used to select a central word in a sentence? Additionally, how many central words are required for a news article? Does this depend on the number of sentences or the overall length of the news?
- In Figure 2, the three (+) symbols are used differently: two in the Multi-hop Syntax Aware Module (Equation 4) and the Semantic Aware Module (Equation 6) signify addition, while the one in the Fake News Detector indicates embedding concatenation (line 500). It's recommended to use distinct symbols for these different operations to prevent confusion.
- Please recheck the notation in Figure 2 (top right corner) to confirm if the 'm_R' in the Syntax Aware Module should be 'm_G' as shown in  Equation 4. If not, an explanation of what 'm_G' represents would be helpful.
- There's a discrepancy in terminology between Figure 2, which mentions a Sequence Aware Module, and the Methods section, which refers to a Semantic Aware Module (4.3). Consistency in terminology is advised.
- The use of only two datasets (Weibo and GossipCop) in this study may limit its generalizability.
- In the ablation study, could an additional experiment be conducted: an ensemble of the graph attention model and the transformer-based model?

**Ethics Review Description:**

There are no ethics issues for this work.

**Reviewer Confidence:**

3: The reviewer is confident but not certain that the evaluation is correct

**Scope:**

3: The work is somewhat relevant to the Web and to the track, and is of narrow interest to a sub-community

---

### Official Review · Reviewer_s76Y · 2023-11-23

**Novelty:** 3
**Technical Quality:** 4

**Review:**

This paper proposes a multi-hop syntax aware fake news detection method MSynFD, which incorporates complementary syntax information to deal with fake news with subtle twists.

Pros: This paper proposes a useful method, MSynFD to handle the complex, subtle twists1 in news articles. The effectiveness of this method is substantiated through empirical validation.

Cons:

1. This paper lacks a discussion on graph-based semantic enhancement methods. In fact, even when narrowing the scope to the fake news detection task, similar methods still exist:
Xu, Weizhi, et al. "Evidence-aware fake news detection with graph neural networks." Proceedings of the ACM Web Conference 2022. 2022.
2. The Keywords Debiasing module exhibits a certain degree of disconnection with other modules of MSynFD. The relevance between this module and the core motivation of the present study appears to be weak.
3. The motivation of this paper and the proposed MSynFD method seem applicable to all NLU tasks requiring fine-grained semantic comprehension, and the relevance to the fake news detection task seems to be weak.

**Questions:**

Apart from the paper mentioned in Cons 2, there is another parallel work closely related to this paper. What differences do you perceive between this paper and MSynFD?

Chen, Zhendong, et al. "A syntactic multi-level interaction network for rumor detection." *Neural Computing and Applications* (2023): 1-14.

**Reviewer Confidence:**

3: The reviewer is confident but not certain that the evaluation is correct

**Scope:**

3: The work is somewhat relevant to the Web and to the track, and is of narrow interest to a sub-community

---

### Official Review · Reviewer_168y · 2023-12-01

**Novelty:** 6
**Technical Quality:** 6

**Review:**

This paper presents a new approach for fake news detection that uses graph neural networks to make use of syntax information in order to catch twists in the framing of the argument. The authors implement this technique and connect it with a transformer model and show that it improves upon the prior state-of-the-art on two different fake news detection datasets, one in Chinese (Weibo) and one in English (GossipCop). They also present ablation studies and representative examples from the datasets to help analyze the model's performance.

This is an interesting paper that makes a useful contribution to the detection of fake news in situations where the exact stance of the article is difficult to understand due to subtle changes in wording.

The approach is well-motivated and documented in detail, which aids its replicability.

The ablation analyses and case studies presented are useful in understanding which factors affect the system’s performance.

However, the performance gains achieved are marginal (~1%) over prior approaches.

It would be helpful to break these results down by the genre of the fake news articles to better understand if there are specific types that this model is better suited for.

The paper presentation is of high quality and the figures are very well made.

---

I have read and replied to the authors' response and made the necessary changes to my review.

**Questions:**

Do the authors have any intuition as to why the optimal number of hops for the two datasets is 4 and 3? Does it relate only to the average length of their examples or also the language- and domain-specific phrase structure?

Typos and presentation improvements:

Lines 620 - 224: future tense is used to describe the experimental setup but it should be past tense instead.

Line 714: “... mitigate the [prior] bias from …”

Line 800 - 870: the style of quotation marks should be standardized.

**Reviewer Confidence:**

3: The reviewer is confident but not certain that the evaluation is correct

**Scope:**

4: The work is relevant to the Web and to the track, and is of broad interest to the community

---

### Decision · Program_Chairs · 2024-01-22

**Decision:**

Accept

**Comment:**

# Strengths:
 * Innovative Approach: The use of a graph neural network to leverage syntax information for detecting subtleties in fake news is innovative.
 * Effective Integration: The paper effectively integrates a syntactical dependency tree with a transformer model, showcasing improvements in fake news detection.
 * Detailed Documentation: The approach is well-documented, aiding replicability.
 * Empirical Validation: The effectiveness of the MSynFD method is supported by empirical validation on two different datasets.
 * Quality Presentation: The paper is well-structured, with clear illustrations and coherent writing.

 # Weaknesses:
 * Marginal Performance Gains: The performance improvement over prior methods is marginal (about 1%).
 * Lack of Genre Analysis: The paper could benefit from a breakdown of results by the genre of fake news.
 * Limited Discussion on Related Work: Some reviewers noted a lack of discussion on graph-based semantic enhancement methods, specifically in the context of fake news detection.
 * Perceived Disconnect in Modules: The relevance and integration of the Keywords Debiasing module with other parts of the MSynFD were questioned.
 * Relevance to Fake News Detection: The relevance of the proposed method to the specific task of fake news detection, as opposed to general NLU tasks, was not clear.

 # Overall:
 The paper presents a novel and useful contribution to the field of fake news detection, particularly in handling subtle linguistic nuances in news articles. While the approach is well-motivated and empirically validated, the paper could be strengthened by addressing the weaknesses noted by the reviewers, such as marginal performance gains, lack of genre-specific analysis, and clearer integration and relevance of different model components. The paper would also benefit from addressing the reviewers' questions in future revisions to enhance clarity and understanding.